# Plant Defense Responses to a Novel Plant Elicitor Candidate LY5-24-2

**DOI:** 10.3390/ijms23105348

**Published:** 2022-05-11

**Authors:** Xin Qi, Kun Li, Lei Chen, Yue Zhang, Nailou Zhang, Wei Gao, Yuedong Li, Xingzhong Liu, Zhijin Fan

**Affiliations:** 1State Key Laboratory of Elemento-Organic Chemistry, College of Chemistry, Nankai University, Tianjin 300071, China; 13121253699@163.com (X.Q.); 18266052144@163.com (K.L.); cl21918@126.com (L.C.); 1120210477@mail.nankai.edu.cn (Y.Z.); gaow@mail.nankai.edu.cn (W.G.); lyuedong2021@126.com (Y.L.); 2Frontiers Science Center for New Organic Matter, College of Chemistry, Nankai University, Tianjin 300071, China; 3State Key Laboratory of Virology, Wuhan Institute of Virology, Chinese Academy of Sciences, Wuhan 430000, China; zhangnailou@wh.iov.cn; 4Department of Microbiology, College of Life Science, Nankai University, Tianjin 300071, China

**Keywords:** plant elicitor, LY5-24-2, immune-inducing activity, systemic acquired resistance, SA signaling pathway

## Abstract

Plant elicitors enhance plant defense against pathogen attacks by inducing systemic acquired resistance (SAR) with no or low direct fungicidal activity. Here we report the synthesis of a novel plant elicitor candidate LY5-24-2 [3,4-dichloro-*N*-(3-chloro-5-(trifluoromethyl)pyridin-2-yl)isothiazole-5-carboxamide] and evaluation of its SAR inducing activity. Bioassays indicated that LY5-24-2 did not show significant anti-fungal activity but provided long-lasting resistance in *Arabidopsis thaliana* (*A. thaliana*) through promoting the accumulation of lignin, cellulose and pectin by 60.1%, 82.4% and 305.6%, respectively, at a concentration of 100 µM. LY5-24-2 also facilitated the closure of leaf stomata and increased the intracellular free Ca^2+^ by 47.8%, induced reactive oxygen species (ROS) accumulation, and inhibited the activity of ascorbate peroxidase (APX, EC 1.11.1.11) and catalase (CAT, EC 1.11.1.6) by 38.9% and 34.0%, respectively, as compared with the control at a concentration of 100 µM. LY5-24-2 induced SAR in plants and was dependent on the *NPR1*-mediated SA pathway by up-regulating expression of 2273 genes in *A. thaliana*. Meanwhile, LY5-24-2 also improved cucumber (*Cucumis sativus* L.) defense against *Pseudoperonospora cubensis* (*P. cubensis*) through promoting ROS accumulation and inhibiting activity of APX and CAT by 30.7% and 23.1%, respectively. Its expression of SA signaling genes *CsNPR1*, *CsPR4* and *CsPR5* was enhanced by 10.8, 5.8 and 6.6 times, respectively. These results demonstrated that LY5-24-2 is a novel elicitor candidate for plant protection via inducing SAR.

## 1. Introduction

Plants are consistently exposed to different abiotic (e.g., draught, temperature) and biotic (e.g., plant pathogens) stresses and their innate immune system helps them to defend themselves against such stresses [1]. Plant cell membrane receptors and intracellular receptors are used to sense pathogens to enable plants to respond quickly to the pathogen invasion, and lead to changes in calcium ions, reactive oxygen species (ROS) and phytohormones [2,3]. Salicylic acid (SA), one of the plant defense hormones, is closely related to resistance against biotrophic and hemi-biotrophic pathogens [4]. In SA signal transduction, *NPR1*, *NPR3*, and *NPR4* are important components involved in two parallel signaling pathways: *NPR1* acts as a transcriptional activator, and *NPR3* and *NPR4* act as transcriptional repressors [5,6]. When SA level is low, the *NPR1* gene is inactive, and *NPR3* and *NPR4* repress the expression of defense genes by interacting with TGA-transcription factors. In contrast, when SA level is elevated after pathogen infection, the activity of *NPR3* and *NPR4* is inhibited, and the transcriptional repression of SA-responsive genes is released [5]. On the other hand, the accumulation of SA promotes the transcriptional activation of *NPR1*, which further induces the expression of defense-related genes [5,7].

Effective management of crop pests, including insects, pathogens and viruses, is an important part of improving food production [8]. However, traditional pesticides with variable levels of residue pose potential threats on food security and environmental risks because of their direct actions on target organisms [8]. Plant elicitors are any compounds that can induce plant immune responses by interacting with endogenous plant defense proteins or activating endogenous plant defense signals to trigger plant defense responses [9]. The earliest plant elicitors are polyacrylic acids and their derivatives that activate *PR1* gene expression in tobacco (*Nicotiana tabacum*) and mediate resistance against tobacco mosaic virus (TMV) or tobacco necrosis virus (TNV) [10]. Among them, 2,2-dichloro-3,3-methylcyclopropanecarboxylic acid enhances peroxidase activity as well as phenolic metabolism in rice and improves resistance against rice blast [11]. Subsequently, 3-allyloxy-1,2-benzisothiazole-1,1-dioxide (Probenzazole, PBZ) showed enhancing resistance to rice blast by activating the expression of defense-related enzymes in rice; it has been widely used in agricultural production for 30 years [12]. SA is a natural plant defense hormone, and exogenous application of SA and its derivatives can promote the accumulation of PR1 protein in tobacco and induce resistance to TMV [13]. As an analogue of SA, 2,6-dichloroisonicotinic acid (INA) is also effective in inducing plant systemic acquired resistance (SAR) [14]. Benzo-(1,2,3)-thiadiazole-7-methionine S-methyl (BTH) is another plant elicitor, triggering a defense response in plants [12]. Both INA and BTH induce SA signaling responses that are less toxic to plants and more effective than direct topical SA application [9]. *N*-(3-chloro-4-methylphenyl)-4-methyl-1,2,3-thiadiazole-5-carboxamide (Tiadinil, TDL) is able to control rice blast [15] and tea anthracnose [16]. It induces the expression of disease resistance genes *PR1*, *PR2* and *PR5* in *Arabidopsis thaliana*, under the infestation of *Pseudomonas syringae* pv. *tomato* DC3000 (*Pst* DC3000) [17]. Isotianil (ISO) is able to induce resistance to rice blast at a low dose and shows good efficacy without direct antifungal activity [18]. Other reported plant elicitors include *N*-cyanomethyl-2-chloroisonicotinamide (NCI), 3-chloro-1-methylpyrazole-5-carboxylic acid (CMPA), and so on [19,20,21].

On the basis of discovery and development of novel plant elicitor methiadinil [4-methyl-*N*-(5-methylthiazol-2-yl)-1,2,3-thiadiazole-5-carboxamide] and its registration as plant elicitor for TMV control in China [22,23,24,25], by modification of the 1,2,3-thiadiazole with cheaper 3,4-dichloroisothiazole, a series of 3,4-dichloroisothiazole-5-carboxamides was synthesized. Bioassay results indicated that compound 3,4-dichloro-*N*-(4,6-dimethoxypyrimidin-2-yl)isothiazole-5-carboxamide and 3,4-dichloro-*N*-(5-methylthiazol-2-yl)isothiazole-5-carboxamide exhibited 41.88% and 42.92% of systemic acquired resistance for tobacco against TMV, and that they are active at the same level as that of isotanil with 44.44% of induction activity [26]. To continue our succeeding novel plant elicitor development, here we report the synthesis of the most active novel plant elicitor candidate, 3,4-dichloro-*N*-(3-chloro-5-(trifluoromethyl)pyridin-2-yl)isothiazole-5-carboxamide (LY5-24-2). Mode of action or the mechanism of action is the basis for directing novel pesticide development and its application. As compared with traditional fungicides, unfortunately, the original mode of action or molecular target of plant elicitor is still unknown. Here we used this novel candidate to probe its inducing mechanism by comparison with commercialized positive controls. *Hyaloperonospora arabidopsidis* (*H. arabidopsidis*) is a downy mildew pathogen of *A. thaliana*, and the laboratory model of *Arabidopsis/H. arabidopsidis* played an important role in cloning and characterizing the major resistance genes (*RPP* genes) and defining of their downstream signaling components [27]. *H. arabidopsidis* was used for the immune-inducing activity assay of LY5-24-2. Bioassays indicated that LY5-24-2 did not show significant anti-fungal activity. In *A.*
*thaliana*, LY5-24-2 induced SAR responses including enhancement of lignin, cellulose, pectin, Ca^2+^ and ROS accumulation, and stomatal closure by depending on the *NPR1*-mediated SA transduction pathway to enhance plant defensive capability. LY5-24-2 could induce these defense responses with the similar mechanism of action not only in *A. thaliana* against *H. arabidopsidis* but also in cucumber against *Pseudoperonospora cubensis* (*P. cubensis*).

## 2. Results

### 2.1. Protection of A. thaliana against H. arabidopsidis by LY5-24-2

LY5-24-2 was synthesized by the condensation of 3,4-dichloroisothiazolyl-chloride with 3-chloro-5-(trifluoromethyl) pyridin-2-amine in relatively high yield (Figure 1). Bioassays indicated that LY5-24-2 did not show significant anti-fungal activity, like ISO (Appendix A). To analyze the immune-inducing activity of LY5-24-2, *A. thaliana* leaves were infected with *Hpa* Noco2 at 24 h after a treatment with LY5-24-2, BTH or ISO. Phenotypic analysis of *A. thaliana* leaves showed that sporulation was more obvious in the control than any of the samples treated with LY5-24-2, BTH or ISO (Figure 2A,C and Appendix A). Lactophenol trypan blue (TB) staining revealed a significant reduction in trailing necrosis on leaves treated with LY5-24-2, BTH or ISO when compared to the control (Figure 2B). At 7 d after inoculation with *Hpa* Noco2, it was found that the number of spores on leaves treated with LY5-24-2, BTH and ISO was reduced by 91.8%, 94.8% and 51.5%, respectively, as compared to that of the control (Figure 2C). To analyze the optimal time of immuno-induction by LY5-24-2, *A. thaliana* leaves were inoculated with *Hpa* Noco2 at 24, 36, 48, 60 and 72 h after spraying 100 μM of LY5-24-2, and spore counts were performed at 7 days post inoculation. The number of spores on *A. thaliana* leaves decreased by 92.7%, 82.5%, 81.9%, 81.3% and 75.0% at 24, 36, 48, 60 and 72 h after a LY5-24-2 treatment, respectively (Figure 2D), which indicated that the effect of the compound on plant immunity was detected between 24 h and 72 h after a treatment, i.e., relatively persistent effects. In order to observe a dose-dependent relationship of LY5-24-2 on immune-inducing activity, experimental results with a series of concentrations starting from 5 to 100 µM (Figure 2E) suggested that there was a good dose–response relationship: LY5-24-2 was able to exert immune induction by reducing 38.4% of spores at 5 μΜ, then reached the highest immune induction by decreasing 92.8% of spores at 100 μM. Taken together, LY5-24-2, like BTH, displayed immune induction in *A. thaliana* model against *Hpa* Noco2 and showed relatively persistent effects.

### 2.2. Enhancement of Accumulation of Cell Wall Components and Facilitation of Stomatal Closure by LY5-24-2

The cell wall, a natural physical barrier of plants, is an important defensive line against pathogens. To investigate the mechanism of action of LY5-24-2, the effect of LY5-24-2 on the cell wall components of *A. thaliana* was studied. LY5-24-2 increased the lignin content of *A. thaliana* leaves by 60.1%, cellulose content by 82.4%, and pectin content by 305.6% (Figure 3A–C). It was found that most of stomata on the leaf of *A. thaliana* were closed after a LY5-24-2 treatment (Figure 3D). Measurements of stomatal length and width showed that stomatal length decreased by 45.7% and stomatal width decreased by 51.7% after a LY5-24-2 treatment (Figure 3E,F). In summary, LY5-24-2 treatments increased the thickness of the cell wall and promoted the closure of stomata, and thus enhanced resistance of *A. thaliana* against pathogen infestation.

### 2.3. Enhancement of Transduction of Intracellular Defense Signals by LY5-24-2

In plants, reactive oxygen species (ROS) are rapidly produced following pathogen infection or physical damages. APX and CAT play a role in scavenging H_2_O_2_. The effects of LY5-24-2 on the activity of APX and CAT enzymes in *A. thaliana* were studied. The results showed that LY5-24-2 inhibited the activity of APX and CAT by decreases of 38.9% and 34.0%, respectively, as compared with the control (Figure 4C,D). The decrease of APX and CAT activity was beneficial to the accumulation of H_2_O_2_; the DAB assay showed that LY5-24-2 promoted the accumulation of ROS in *A. thaliana* leaves (Appendix A). Furthermore, the results showed an enhancement of Ca^2+^ fluorescence signal in cells after a LY5-24-2 treatment. The measurement results showed that the fluorescence intensity of cells treated with 10 µM of LY5-24-2 increased by 47.8% (Figure 4A,B), indicating that LY5-24-2 promoted the accumulation of Ca^2+^ in *A. thaliana* cells. H_2_O_2_ and Ca^2+^ are important components of the plant immune system and essential intracellular defense signaling substances in plants. It is proposed that LY5-24-2 can activate plant immune responses and improve plant defense ability by promoting the accumulation of ROS and Ca^2+^.

### 2.4. Analyses of Differentially Expressed Genes (DEGs) in A. thaliana after a LY5-24-2 Treatment

To elucidate a possible regulatory network of LY5-24-2, RNA-Seq was performed with LY5-24-2-treated and control samples of *A. thaliana*. Firstly, the gene expression profiles of LY5-24-2-treated and control were compared, and a total of 3802 DEGs were identified (Figure 5A). A total of 2273 DEGs were up-regulated, while 1529 were down-regulated by LY5-24-2, indicating that the expression of a large number of genes was changed after a LY5-24-2 treatment (Figure 5A). To identify the molecular pathways of specific LY5-24-2-responsive genes, gene ontology (GO) analysis was performed with the DEGs, and 30 enriched GO terms were identified in LY5-24-2-treated plants, which were significantly enriched (Figure 5B). Among these 30 enriched GO terms, 28 enriched GO terms were associated with biological processes, 1 enriched GO term was associated with cellular components, and 1 enriched GO term was associated with molecular functions. These suggested that LY5-24-2 might have the most significant effects on genes related to biological processes. To better understand the cellular pathways involved in the up—regulation of DEGs by LY5-24-2, KEGG pathway analysis was performed. The results suggested that these DEGs were involved in endoplasmic reticulum protein processing, purine metabolism, plant–pathogen interactions and amino acid biosynthesis processes (Figure 6). RNA-Seq analysis showed that a LY5-24-2 treatment triggered a high level of changes in differential gene expression in *A. thaliana*, including genes related to plant–pathogen interactions, which might provide an insight into the mechanism of LY5-24-2-induced immune action.

### 2.5. Mode of Action of LY5-24-2 on Downstream SA Signal Transduction and Dependence on AtNPR1

SA is an important plant defense hormone that promotes immunity against pathogens. To determine the immune induction mode of action of LY5-24-2, real-time qPCR analysis was performed with LY5-24-2-treated *A. thaliana*, and genes involved in the SA signaling pathway such as *AtNPR1*, *AtPR1*, *AtPR2* and *AtPR3* were included. The results showed that *AtNPR1*, *AtPR1*, *AtPR2* and *AtPR3* genes were up-regulated after a LY5-24-2 treatment by 3.7, 2.9, 3.6 and 3.5 times, respectively (Figure 7A), suggesting that LY5-24-2 was involved in the SA signaling pathway. In order to identify the key components of LY5-24-2 which exert an immune induction function, mutants of key genes of SA signaling pathways including *npr1*, *tga3* and *tga7* were selected for LY5-24-2 immune induction assays in this study. *npr1*, *tga3* and *tga7* were verified by PCR (Appendix A). By monitoring the number of spores at 7 days post inoculation after inoculation with *Hpa* Noco2, LY5-24-2 was found to be effective in resistance to *Hpa* Noco2 in the wild-type (Figure 7B and Appendix A). Compared with the control, the number of spores of wild-type was reduced by 90.9% after a LY5-24-2 treatment. In the *npr1*, compared to the control, spores were reduced by 18.5% after a LY5-24-2 treatment (Figure 7B). In *tga3* and *tga7*, by comparison with the control, the spores were reduced by 48.6% and 66.9% after a LY5-24-2 treatment, respectively (Figure 7B). These results suggested that loss of *AtNPR1* gene caused a severe decrease in the immune-inducing activity of LY5-24-2, and loss functions of *AtTGA3* and *AtTGA7* had a relatively limited reduction in the immune-inducing activity of LY5-24-2. It was concluded that LY5-24-2 was able to induce the expression of SA signaling pathway genes and depended on the *AtNPR1*-mediated SA signaling pathway to play the immune-inducing function in *A. thaliana.*

### 2.6. Defense Enhancement of Cucumber against P. cubensis by LY5-24-2

Experimental data described above showed that LY5-24-2 enhanced the resistance of *A. thaliana.* against *Hpa* Noco2 infection, accumulated intracellular Ca^2+^, and acted in the *AtNPR1*-mediated SA signaling pathway. To evaluate whether LY5-24-2 induces SAR in other plants, further experiments were carried out with a cucumber (*Cucumis sativus* L.) system. Cucumber leaves were infected with *P. cubensis* at 24 h after spraying with LY5-24-2, BTH or ISO. The leaf phenotypic analysis showed that the plaque of the leaves treated with LY5-24-2 was significantly less than that of the control, which was similar to that of BTH and ISO (Figure 8A). In cucumber, the gene expression of *CsNPR1*, *CsPR4*, *CsPR5, CsAPX* and *CsCAT* was analyzed in leaves treated with LY5-24-2. A LY5-24-2 treatment promoted the expression of *CsNPR1*, *CsPR4*, *CsPR5* and inhibited the expression of *CsAPX* and *CsCAT*. The expression of *CsNPR1*, *CsPR4*, *CsPR5* was up-regulated by 10.8, 5.8 and 6.6 times, respectively (Figure 8B), and the expression of *CsAPX* and *CsCAT* was down regulated by 0.6 and 0.4 times, respectively (Figure 8C). Further analysis of APX and CAT enzyme activity in LY5-24-2-treated cucumber leaves showed that LY5-24-2 decreased APX and CAT activity by 30.7 and 23.1%, respectively, after a LY5-24-2 treatment (Figure 8E,F). In addition, DAB staining results showed that LY5-24-2 promoted the accumulation of ROS in cucumber leaves (Figure 8D). In cucumber, LY5-24-2 improved the defense response against *P. cubensis*, just like the mode of action in *A. thaliana.* The changes in *PR* gene expression, APX and CAT enzyme activity were also consistent with those results obtained from *A. thaliana.* In cucumber, LY5-24-2 also promoted *PR* gene expression and inhibited APX and CAT enzyme activity. These suggested that LY5-24-2 could induce defense responses in a variety of plants with the same mechanism of action.

## 3. Discussion

Plant elicitors are regarded as new plant protection agents in the 21st century. On the basis of our previous development of plant elicitor methiadinil [24], herein, LY5-24-2 was discovered and demonstrated as a plant defense elicitor candidate to enhance the activity of *A.*
*thaliana* against *Hpa* Noco2 and cucumber against *P.*
*cubensis*. Analyses of the mechanism of action of immune-inducible activity showed that LY5-24-2 increased plant SAR by promoting stomatal closure, thickening the cell wall, inhibiting scavenge hydrogen peroxide enzymes, accumulating Ca^2+^ and ROS and relying on the *NPR1*-mediated transduction process of SA signaling pathway. These results suggested that LY5-24-2 could perform an immune-inducing activity. Plant elicitors are small molecules that induce plant immune responses and can protect plants from diseases by mimicking the interaction of natural inducers or defense signaling molecules with their respective cognate plant receptors, or by interfering with other defense signaling components to trigger defense responses [9]. Plant defense elicitors often do not have direct fungicidal activity, and this will avoid residual toxicity compared with traditional fungicides [28]. Promotion and application of plant defense elicitors will help to solve problems associated with food security and environment.

The majority of plant defense elicitors, such as PBZ, INA, BTH, TDL and ISO, are designed to improve plant defenses by inducing SAR. SA pathway is the main defense signaling pathway on which most plant defense elicitors depend [9]. BTH and ISO have been commercialized as plant elicitors and are widely used in agriculture production. BTH triggers NPR1 protein-dependent SAR in *A. thaliana* and inhibits APX and CAT function [29,30]. ISO enhances plant defense by inducing the accumulation of defense-related enzymes and the expression of defense-related genes [31]. Like BTH, LY5-24-2 was dependent on NPR1 protein to induce immune responses. In inducing immunity of *A. thaliana* against *H. arabidopsidis*, LY5-24-2 induced the same level of effect as BTH and was superior to ISO. Studies of the mechanism of action of plant elicitors often relies on the immune signaling pathways that have been resolved so far. DCA (3,5-dicholoroanthranilic acid) and BTHC (2-(5-Bromo-2-hydroxy-phenyl)-thiazolidine-4-carboxylic acid) were identified from a library of 114 drug-like organic compounds, which trigger rapid, intense and transient resistance of *A. thaliana* to the pathogenic oomycete Hpa and the bacterial pathogen *Pseudomonas syringae* [32]. In contrast to INA- and BTH-mediated immunity, which are exclusively dependent on the transcriptional cofactor and SA coreceptor *NPR1* [9], DCA-mediated immunity is only weakly dependent on *NPR1* and partially dependent on the signaling pathway of the *WRKY70* transcription factor [30]. Floro-pyrazolo [3,4-*d*]pyrimidine derivative as a novel plant activator induces SAR and induced systemic resistance (ISR) pathways that can highly induce resistance to plant pathogens [33]. In this study, LY5-24-2, as a plant immune activator candidate, was identified as partially dependent on the *NPR1*-mediated SA signaling pathway to enhance resistance to Hpa in *Arabidopsis*. Knowledge about the targets of plant defense elicitors is very limited. The discovery of new targets and new leads is a key scientific issue in the novel pesticide development. Future research on the mechanism of action of plant immune activators will be the focus of target identification of plant activators too.

## 4. Materials and Methods

### 4.1. Synthesis of LY5-24-2

All reagents and ultra-dry solvents were purchased from Bide Pharmatech Ltd. (Shanghai, China), Infinity Scientific (Beijing) Co., Ltd. (Beijing, China), Heowns Biochem LLC (Tianjin, China), Energy Chemical (Shanghai, China), or Meryer (Shanghai) Chemical Technology Co., Ltd. (Shanghai, China). Other analytical solvents were purchased from Tianjin Sixth Chemical Reagent Factory (Tianjin, China) or Tianjin Bohua Chemical Reagents Co., Ltd. (Tianjin, China).

Melting points were measured on an X-4 Digital Type Melting Point Tester (Henan Gongyi Yingyi Yuhua Instrument Co., Ltd., Gongyi, Henan), and data reported were uncorrected. ^1^H NMR (400 MHz), ^13^C NMR (101 MHz) and ^19^F NMR (376 MHz) spectra were recorded on a Bruker Avance 400 MHz spectrometer (Billerica, MA, USA) in CDCl_3_ solution with tetramethylsilane as an internal standard. High-resolution mass spectra (HRMS) were recorded with an Agilent 6520 Q-TOF LC/MS instrument (Agilent Technologies Inc., Santa Clara, CA, USA).

3,4-Dichloroisothiazole-5-carbonyl chloride was synthesized according to previously reported procedures (Figure 1). To a 100 mL round-bottom flask, thionyl chloride (30 mL, 0.41 mol) was added to 3,4-dichloroisothiazole-5-carboxylic acid (10.0 g, 50 mmol). The mixture was refluxed for 5 h, then the thionyl chloride was removed by distillation. Compound **2** (10.6 g, yield of 96%) was obtained by distillation under vacuum (b.p.: 80–86 °C/1.5–2.0 mbar), which could be solidified into white crystals at room temperature.

To synthesis of LY5-24-2 (3,4-dichloro-*N*-(3-chloro-5-(trifluoromethyl)pyridin-2-yl)isothiazole-5-carboxamide), 3-chloro-5-(trifluoromethyl)pyridin-2-amine (0.39 g, 2.0 mmol), 4-DMAP (0.30 g, 2.5 mmol) and dichloromethane (DCM, 10 mL) were added to a 100 mL round-bottom flask (Figure 1). A solution of 3,4-dichloroisothiazolecarbonyl chloride in DCM was added dropwise at room temperature. The mixture was stirred at room temperature overnight. After the reaction was completed, the reaction mixture was diluted with DCM (15 mL) and water (10 mL). The organic layer was separated, and the aqueous layer was extracted with DCM (2 × 10 mL). The combined organic solution was washed sequentially with brine (2 × 10 mL). The resultant organic layer was dried over anhydrous Na_2_SO_4_. The solvent was evaporated under reduced pressure, which was then purified by column chromatography on a silica gel (100–200 mesh) with a mixture of ethyl acetate/petroleum ether (60–90 °C fraction) (1:10–1:2, *v*/*v*) to give LY5-24-2.

White powder; yield: 78%; m.p.:173–174 °C; ^1^H NMR (400 MHz, Chloroform-*d*) δ 9.41 (s, 1H), 8.74 (s, 1H), 8.08 (s, 1H). ^13^C NMR (101 MHz, Chloroform-*d*) δ 156.34, 154.57, 149.46, 149.26, 144.19, 135.48 (q, *J* = 3.2 Hz), 124.97, 124.63, 122.37 (q, *J* = 272.7 Hz), 121.48, 119.28.

### 4.2. Plant Materials, Growth Conditions, Treatments with Compound Solutions and Pathogen Inoculation

All *A. thaliana* used in this study were two-week old Columbia (Col-0). A mutant with *npr1* used in this study was previously described by Zheng [34]. NPR1 acts as a receptor in SA transduction. TGA3 and TGA7 encode bZIP transcription factor family protein and are involved in the SA signaling pathway. Mutants of *t**ga3* (SALK_088114C) and *tga7*(SALK_114488.1) were ordered from Arahsare (https://www.arashare.cn/index.html, accessed on 9 September 2020).

For analysis of LY5-24-2 immune-inducing activity and mechanism of action, *A. thaliana* (Col-0, *npr1*, *tga3*, *tga7*) seeds were sown in soil in a growth chamber on a 16 h light (22 °C)/8 h dark (20 °C) cycle for 2 weeks. Twenty seedlings of *A. thaliana* were sprayed with 1 mL of 100 µM of LY5-24-2, BTH or ISO, respectively; 0.2% DMF (*N*,*N*-Dimethylformamide) without test compound was used as control [35]. Twenty-four hours after the chemical treatment, seedlings were inoculated with *H. arabidopsidis* isolate Noco2 with 5 × 10^4^ spores/mL according to the description of Mcdowell et al. [36]. After inoculation, seedling melons were placed in an incubator at 18 °C with 80–100% humidity and were monitored for 7 days. To determine the number of spores, collected seedlings were vortexed in sterile water and counted in a hemocytometer [32]. In addition, a series of concentrations of LY5-24-2, including 5, 10, 20, 40, 60, 80 and 100 μΜ, were used to explore a dose–response relationship and different time points including 24, 36, 48, 60 and 72 h were used to find out the persistence of its immune-inducing activity at a concentration of 100 μΜ. For analysis of LY5-24-2 immune-inducing activity in *Cucumis sativus* L. (cucumber) (Xintai Mi Ci, a typical cucumber of northern China), the cucumber seeds were sown into the soil, then grown in a chamber on a 16 h light (25 °C)/8 h dark (23 °C) cycle for 2 weeks. The cucumber seedlings were sprayed with 0.2% DMF (control), 100 µM of LY5-24-2, BTH or ISO, respectively, 24 h before inoculation. The leaves of the cucumber were dipped into a *P. cubensis* spore suspension (1 × 10^5^ spores/mL). After inoculation, cucumber seedlings were placed in an incubator with 25 °C and 80–100% humidity. Disease development was assessed 7 d after inoculation.

### 4.3. Lactophenol Trypan Blue Staining

The lactophenol trypan blue (TB) staining solution was composed of 10 mL phenol, 10 mL glycerol, 10 mL lactic acid, 10 mL H_2_O_2_, and 10 mg of Taipan blue dye. *A. thaliana* leaves infested with *H. arabidopsidis* isolate Noco2 were decolorized by chloral hydrate after boiling in TB staining solution for 5 min.

### 4.4. Cell Wall Component Measurements

*A. thaliana* leaves were assayed for lignin content after 24 h of treatment with 1 mL of 100 μΜ LY5-24-2. The leaves were collected and dried in an oven at 80 °C. The dried leaves were crushed and sieved through mesh screen, and 5.0 mg of leaf powder was used for lignin content measurement according to the instructions of the Lignin Content Assay Kit (Solarbio Science & Technology Co., Ltd., Beijing, China).

*A. thaliana* leaves were used for cellulose content measurement after 24 h of treatment with 1 mL of 100 μΜ LY5-24-2. Weighing 0.3 g of leaves to freeze and grind in liquid nitrogen, the cellulose content was analyzed according to the instructions of the Cellulose Content Assay Kit (Solarbio Science & Technology Co., Ltd., Beijing, China).

*A. thaliana* leaves were used for total pectin content measurement after 24 h of treatment with 1 mL of 100 μΜ LY5-24-2. Weighing 50 mg of leaves to freeze and grind in liquid nitrogen, the pectin content was analyzed according to the instructions of the Total Pectin Assay Kit (Solarbio Science & Technology Co., Ltd., Beijing, China).

### 4.5. Stomatal Morphology of A. thaliana Leaves

*A. thaliana* seedlings were used to observe variation of stomatal morphology after 24 h of treatment with 1 mL of 100 μΜ LY5-24-2. The leaves were fixed, dehydrated, dried and plated with Pt. The leaves were observed by scanning electron microscopy (QUANTA 200, FEI Company, Hillsboro, OR, USA). The length and width of stomata were determined by Image J software.

### 4.6. Analysis of Ca^2+^ Content

*A. thaliana* seedlings were prepared for protoplasts analysis according to the method of Yoo et al. [37]. When the protoplasts were treated with 100 µM LY-5-24-2, the protoplasts were heavily fragmented, so the LY-5-24-2 treatment concentration was lowered to 10 µM. We measured 200 μL of the protoplast suspension with termination concentration of 10 μΜ LY5-24-2 and 5 μΜ intracellular calcium fluorescence probe Flou-4AM (CAS: 273221-67-3) for Ca^2+^ content analysis after incubation at 25 °C for 30 min. The blank control was treated with the same amount of solvent DMF and Flou-4AM. The confocal microscope (Zeiss LSM700, CarlZeiss, Oberkochen, Germany) was used for visualizing the fluorescence in protoplast cells. ZEN 2010 software (CarlZeiss, Oberkochen, Germany) was used for image acquiring. Fluorescence intensity was determined by Image J software.

### 4.7. Transcriptome Analysis

*A. thaliana* leaves, treated with 100 μΜ of LY5-24-2 after 24 h, were used for RNA-Seq in this transcriptomic study. RNA-Seq analysis was performed at Novogene Bioinformatics Technology Co., Ltd. (Tianjin, China). Differential expression analysis of RNA-Seq was performed using limma software. The differentially expressed genes induced by LY5-24-2 were selected for GO and KEGG analyses on the Geneontology (http://geneontology.org/, accessed on 12 January 2020) and KEGGPATHWAY databases (https://www.kegg.jp/kegg/pathway.html, accessed on 12 January 2020), respectively [25].

### 4.8. RNA Extraction and Quantitative Real-Time PCR

Total RNA of *A. thaliana* seedlings, treated with 100 μΜ of LY5-24-2 after 24 h, was extracted by using the Plant RNA Extraction Kit (TianMo biotech, China). cDNA was synthesized according to the instructions of PrimeScript™ RT reagent Kit with gDNA Eraser (TAKARA, Japan). Real-time PCR amplification was performed using the TB Green Premix Ex Taq (Tli RNaseH Plus) (TAKARA, Japan) and a fluorescence quantitative PCR instrument (Bio-rad CFX96, Hercules, CA, USA). Relative gene expression in different samples was calculated using the 2^−∆∆Ct^ method, using Ct values for each sample with a specific fluorescence threshold [38]. The *AtTUB4* of *A. thaliana* and *CsActin* of cucumber were used as internal controls. Quantitative real-time PCR was performed in triplicate. The primers used for quantitative real-time PCR are listed in Appendix A.

### 4.9. Measurement of AXP and CAT Activity

*A. thaliana* leaves were used for measurement of APX (ascorbate peroxidase APX, EC 1.11.1.11) and CAT (catalase, EC 1.11.1.6) activity after treatment with 100 μΜ of LY5-24-2 for 24 h. Weighing 0.1 g of leaves to freeze and grind in liquid nitrogen, AXP and CAT activity was analyzed according to the instructions of the Ascorbate Peroxidase (APX) Activity Assay Kit and Catalase (CAT) kit (Suzhou Comin Biotechnology Co., Ltd., Suzhou, China).

### 4.10. Measurement of ROS Accumulation

*A. thaliana* leaves, treated with 100 μΜ of LY5-24-2 after 24 h, were used for ROS measurement by DAB (3,3-diaminobenzidine) staining according to the description of Zheng et al. [34].

### 4.11. Data Analysis

Student’s *t*-test was used to calculate difference*s*. GraphPad Prism 8.1 and Statistix 8.1 were used for the data analysis.

## Figures and Tables

**Figure 1 ijms-23-05348-f001:**
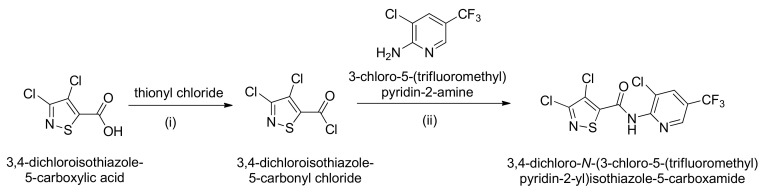
Synthetic route of the compound LY5-24-2. Reagent and conditions: (i) SOCl_2_, reflux, 5 h. (ii) 4-DMAP, CH_2_Cl2, CH2Cl2, r.t., overnight.

**Figure 2 ijms-23-05348-f002:**
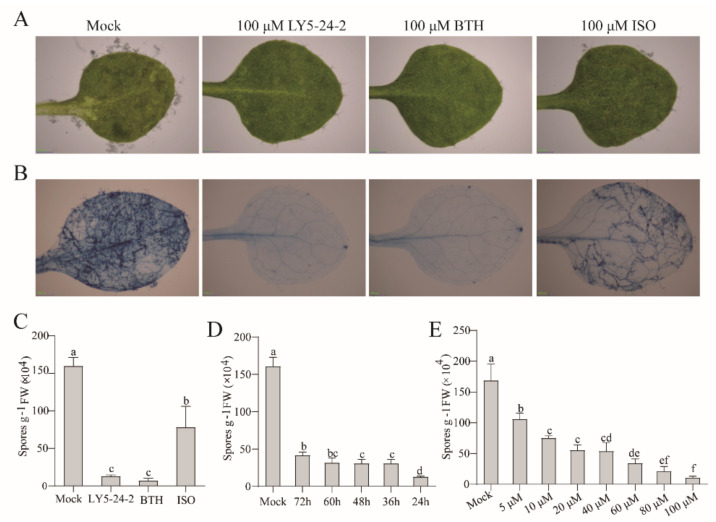
LY5-24-2 enhanced resistance of *A. thaliana* against *H. arabidopsidis* Noco2. (**A**) Phenotypes of leaves of wild-type (WT) *A. thaliana* inoculated with *H. arabidopsidis* Noco2 at 24 h after a treatment with 100 μM of LY5-24-2, BTH or ISO. Leaves were harvested at 7 days post inoculation (dpi), Bars, 500 μm. (**B**) Photographed and stained with lactophenol trypan blue (TB). Bars, 500 μm. (**C**) Sporulation level of *H. arabidopsidis* Noco2 of WT, spraying-inoculated with *H. arabidopsidis* Noco2 at 24 h after a treatment with 100 μM of LY5-24-2, BTH or ISO. (**D**) Sporulation level of *H. arabidopsidis* Noco2 of WT, spraying-inoculated with *H. arabidopsidis* Noco2 at 24, 36, 48, 60 and 72 h after a treatment with 100 μM of LY5-24-2. (**E**) Sporulation level of *H. arabidopsidis* Noco2 of WT, spraying-inoculated with *H. arabidopsidis* Noco2 at 24 h after a treatment with 5, 10, 20, 40, 50, 80 or 100 μM of LY5-24-2. Seedlings were harvested at 7 days post inoculation (dpi), and spores were counted. Data are shown as the mean of three biological replicates ± SD. The letters represent significant difference (*p* < 0.05, Student’s *t*-test).

**Figure 3 ijms-23-05348-f003:**
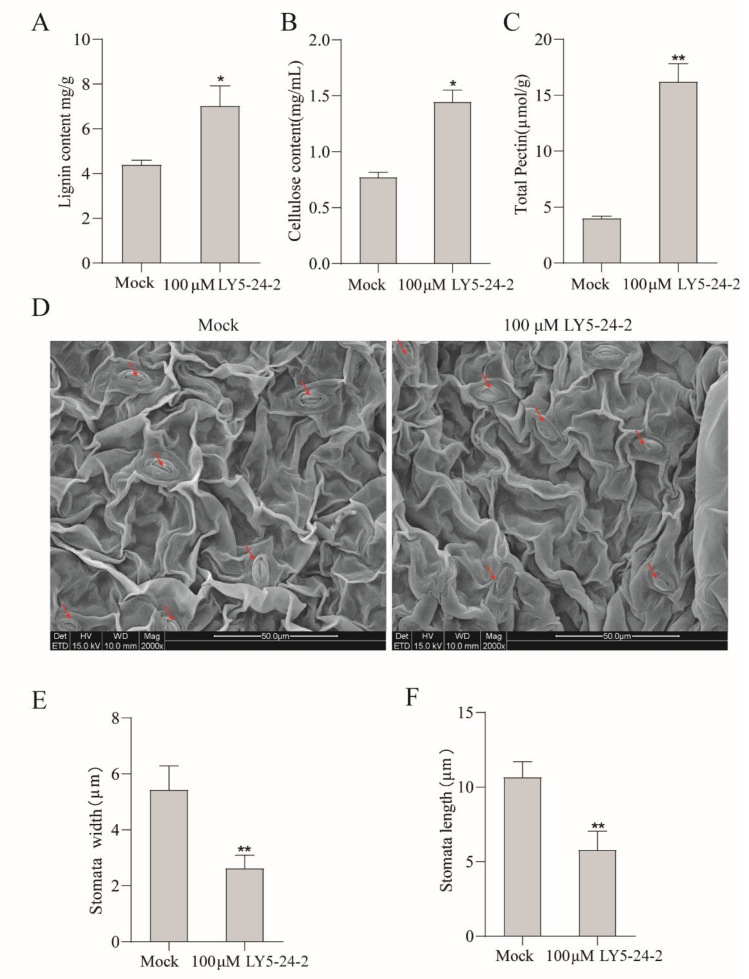
LY5-24-2 promoted the accumulation of *A. thaliana* cell wall components and the closure of stomata at 24 h after a treatment with 100 μM of LY5-24-2. (**A**) Lignin content. (**B**) Cellulose content. (**C**) Total pectin content. (**D**) Changes in stomata (Red arrows indicated leaf stomata). (**E**) Stomatal width. (**F**) Stomatal Length. Data are shown as the mean of three biological replicates ± SD. The * represent significant difference (* *p* < 0.05, ** *p* < 0.01, Student’s *t*-test).

**Figure 4 ijms-23-05348-f004:**
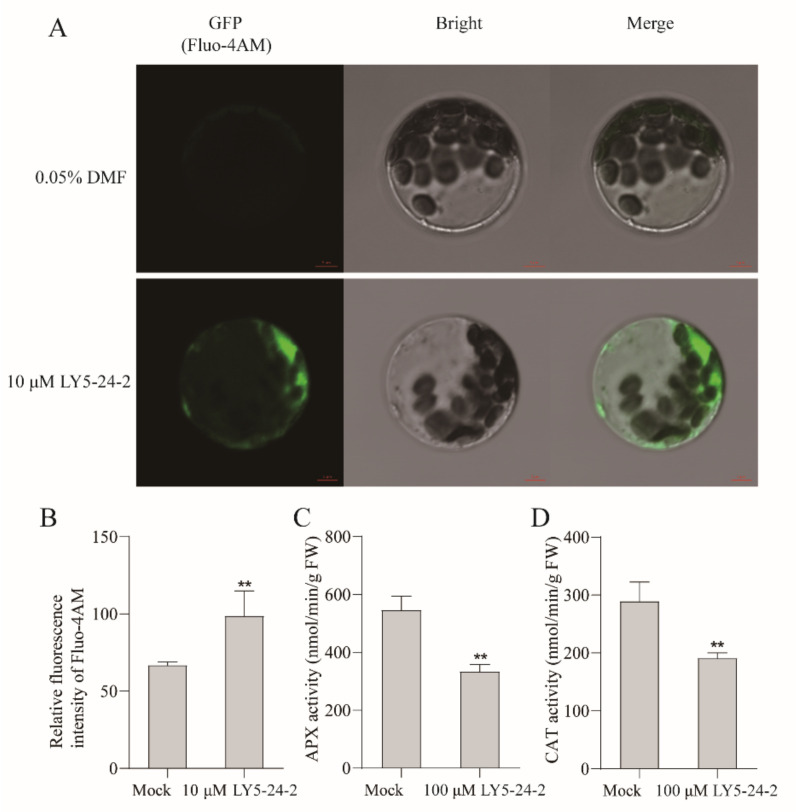
LY5-24-2 promoted the accumulation of Ca^2+^, inhibited APX and CAT activity in *A. thaliana* at 24 h after a treatment with 100 μM of LY5-24-2. (**A**) Ca^2+^ content analysis. (**B**) Relative fluorescence intensity statistics of A. (**C**) APX activity. (**D**) CAT activity analysis. Data are shown as the mean of three biological replicates ± SD. The * represent significant difference (** *p* < 0.01, Student’s *t*-test).

**Figure 5 ijms-23-05348-f005:**
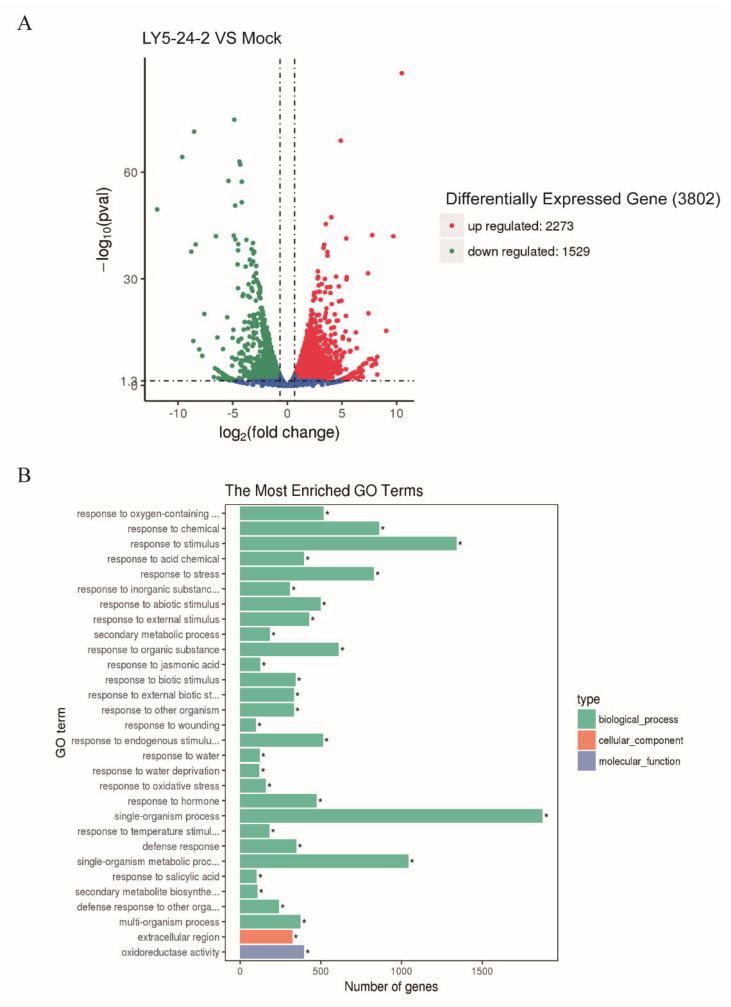
Identification of differentially expressed genes (DEGs) and GO enrichment analysis of *A. thaliana* after a LY5-24-2-treatment. (**A**) Identification of DEGs of *A. thaliana* from the LY5-24-2 treatment and the control (*p* < 0.05). (**B**) GO enrichment analysis of DEGs in *A. thaliana* after a LY5-24-2 treatment (* *p* < 0.05, Student’s *t*-test).

**Figure 6 ijms-23-05348-f006:**
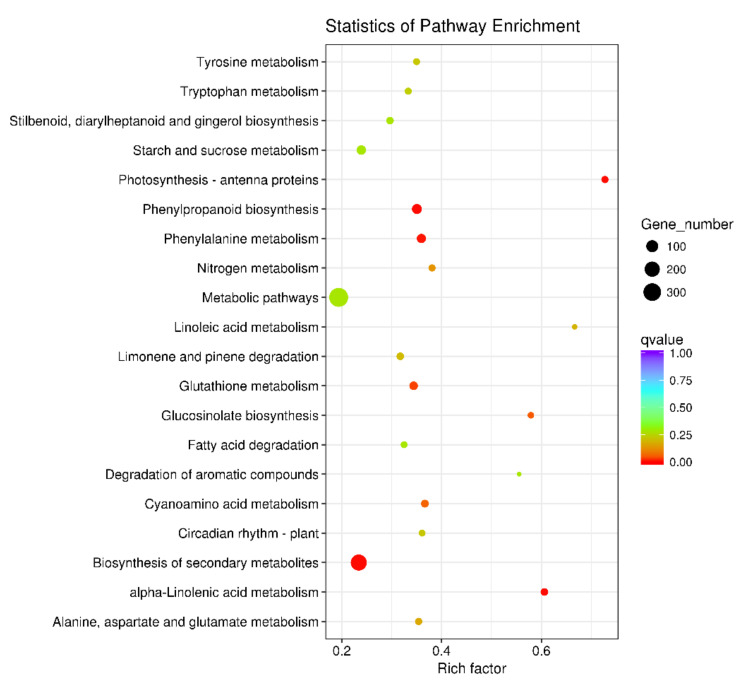
KEGG pathway analysis of up-regulated genes in *A. thaliana* leaves after a LY5-24-2 treatment (*p* < 0.05, Student’s *t*-test).

**Figure 7 ijms-23-05348-f007:**
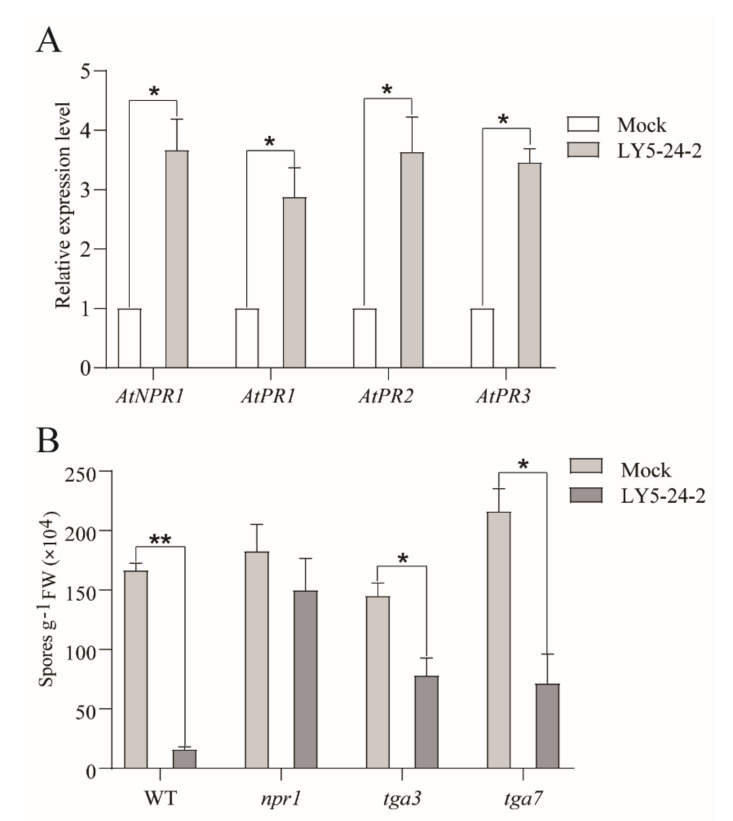
Dependence of LY5-24-2 on SA signaling for immune-inducing functions. (**A**) *AtNPR1*, *AtPR1*, *AtPR2* and *AtPR3* expression analysis in *A. thaliana* leaves at 24 h after a treatment with 100 μM of LY5-24-2. *AtTUB4* was served as an internal control. (**B**) Sporulation level of *H. arabidopsidis* Noco2 on of WT, *npr1*, *tga3* and *tga7* spraying-inoculated with *H. arabidopsidis* Noco2 at 24 h after a treatment with 100 μM of LY5-24-2. Data are shown as the mean of three biological replicates ± SD. The * represent significant difference (* *p* < 0.05, ** *p* < 0.01).

**Figure 8 ijms-23-05348-f008:**
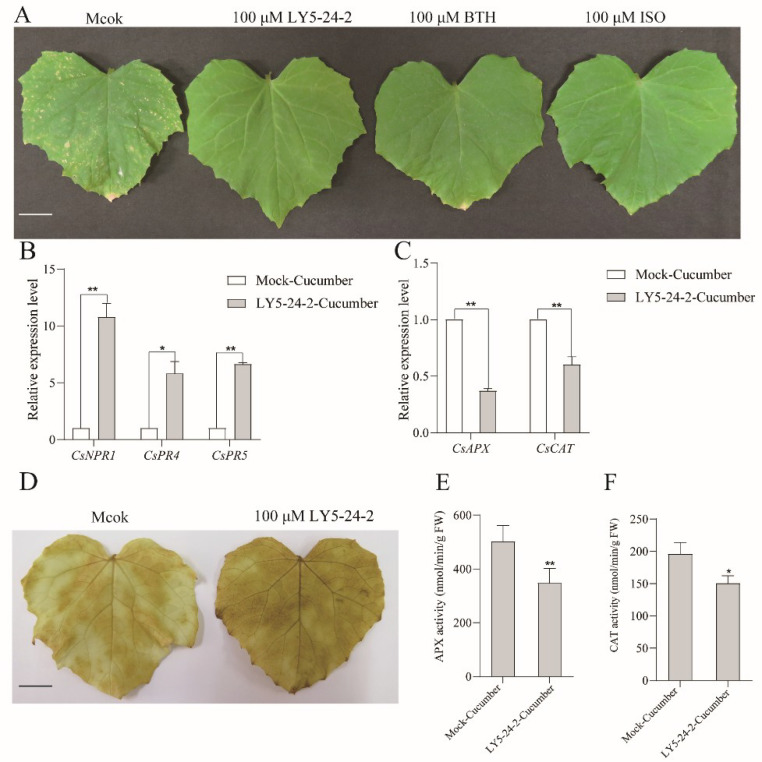
LY5-24-2 promoted resistance of cucumber to *P. cubensis*. (**A**) Phenotypes of leaves of cucumber inoculated with *P. cubensis* at 24 h after a treatment with 100 μM of LY5-24-2, BTH or ISO. The 21-day-old cucumber seedlings were spray-inoculated with *P. cubensis.* Leaves were harvested at 7 dpi. Bars, 2 cm. (**B**) Analysis of *CsNPR1*, *CsPR4* and *CsPR5* expression in cucumber leaves at 24 h after a treatment with 100 μM of LY5-24-2. (**C**) Analysis of *CsAPX* and *CsCAT* expression. *CsActin* was served as an internal control. (**D**) DAB staining in cucumber at 24 h after a treatment with 100 μM of LY5-24-2. Bars, 2 cm. (**E**) Analysis of APX activity in cucumber at 24 h after a treatment with 100 μM of LY5-24-2. (**F**) Analysis of CAT activity. Data are shown as the mean of three biological replicates ± SD. The * represent significant difference (* *p* < 0.05, ** *p* < 0.01, Student’s *t*-test).

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
