# Peer review of "Plant Defense Responses to a Novel Plant Elicitor Candidate LY5-24-2"

_ijms, 2022, doi:10.3390/ijms23105348_

Round 1

Reviewer 1 Report

In this manuscript (Manuscript ID: ijms-1700968) a novel plant elicitor candidate, LY5-24-2 was synthetized. The potential use of LY5-24-2 as an elicitor was tested on Arabidopsis thaliana infected by Hyaloperonospora arabidopsidis, and was found effective. Similarly, 24 h LY5-24-2 treatment alleviated Pseudoperonospora cubensis infection in cucumber (Cucumis sativus). Arabidopsis and cucumber plants were treated with LY5-24-2 and other elicitors, and changes in cell wall components, H2O2 and Ca2+ levels, ascorbate peroxidase and catalase activities, stomatal morphology were investigated. RNA-Seq analyses were performed on LY5-24-2 treated and control Arabidopsis plants too. To elucidate the signaling pathway participating in LY5-24-2 response three Arabidopsis mutant lines (npr1, tga3 and tga7) were used. The results provide evidences of the efficiency of LY5-24-2 in enhancement of defense against the used pathogens, and of the role of NPR1 pathway in the signaling.

The Authors can find several remarks and questions bellow:

English language is fine, however minor spell check is required, and typos within the text need to be corrected. I suggest correcting the final manuscript carefully.

The whole scientific name of the organisms (plants and pathogens) should be given at least when first mentioned and in Materials and methods section.

Check the size of letters in the Abstract.

The Introduction is acceptable. A short paragraph about salicylic acid signal transduction is followed by a detailed description of several well-known elicitors. The authors could discuss shortly the response of Arabidopsis to H. arabidopsidis to help the reader in the understanding of this mechanism and the chosen parameter which were studied. Please reconsider sentence in lines 85-93, since it is too long. In lines 89-90 “…a series 3,4-dichloroisothiazole-5-carboxamides with systemic acquired resistance were…” is written, but it is unclear what that mean.

Materials and methods must be improved, and the Authors should reconsider the structure of it, and reduce the number of subsections.

Synthesis of LY5-24-2 could be described in one subsection, and Figure 1 legend could provide the reader with slightly more details.

Cell wall component measurements can be in a single subsection also.

Description of lactophenol Trypan Blue (TB) staining is missing.

Were the Arabidopsis plants really grown on 20 °C at day and 22 °C at night?

What is the meaning of DMF? Use the whole name of chemicals when first mentioned.

Which statistical method was used to calculate differences? The software and the methods should be given in the Materials and methods.

Plants were sprayed with DMF as mock treatment and with LY5-24-2, ISO and BTH as elicitors. What was the volume of the treatments?

Please, try to avoid using numbers at the beginning of the sentence, when it is possible.

Whole name of ascorbate peroxidase and catalase, and EC numbers of the enzymes should be given.

As I mentioned earlier several typos can be found in the manuscript, but especially in Materials and methods. Some examples can be found bellow:

lines 112, 121,132, and 145: dots are unnecessary after subsection titles. What are the meaning of (2) and (3) after the titles in lines 121 and 132?

line 153: correct “Zheng et al”

lines 155-156: correct tag3 and tag7 to tga3 and tga7

line 181: correct “… 40 mesh…”

Please, reconsider sentence in line 201-202

Results are presented clearly, but some minor changes can be made:

Results from DMF treated plants served as control throughout the manuscript. However the question arises, is there any effect of the 0.2 % DMF treatment on the plants when compared to non-treated plants? Have the Authors measured any parameters in non-treated plants? It can be added as supplementary material to the manuscript, to demonstrate that there were no significant difference between DMF treated and non-treated plants.

It is rather unusual to refer to P-value in the text, it should be avoided.

Supplementary Tables and Figures, and the description of Supplementary material are missing from the manuscript, it should be added. Figure S7 is mentioned in line 357, however no reference to Figures S3-6 can be found in the manuscript, it should be corrected.

The Authors use Hpa Noco2 and Hpa. Noco2 abbreviation as well, it should be unified.

Fig. 2 – scale bars are not enough visible. Check the title of y axes on Fig 4C and 4D.

Rewrite Figure legends of Figs 2, 3, and 4, and correct the sentence to “Data were shown as the mean of three biological replicates ± SD.” on Figure legends 2, 3, 4, 7, and 8.

Reference to Figure 3C and Figure 3D should be added and corrected in line 286 and 287, respectively.

Discussion is quite short, and did not contain any comparison of the results of this study and the known literature about the well described elicitors ISO and BTH. The results of the newly synthetized elicitor LY5-24-2 and data of other elicitors from the literature can be compared, to emphasize its effectiveness.

References should be thoroughly checked also. In line 495 Mcdowell should be written as McDowell and the title should be corrected. Reference in line 493 has two authors, and it should be referred in the text as Livak and Schmittgen, 2001.

In my opinion the manuscript can be accepted after major revision.

Author Response

 Reply to the Review Report

The whole scientific name of the organisms (plants and pathogens) should be given at least when first mentioned and in Materials and methods section.

Answer: Arabidopsis thaliana, Cucumis sativus L., Pseudoperonospora cubensis have been given in Abstract in line 20, 28 and 29.

Arabidopsis thaliana, Hyaloperonospora arabidopsidis, Cucumis sativus L. and Pseudoperonospora cubensis have been given in Materials and methods in line 158, 169, 177 and 182

Check the size of letters in the Abstract.

Answer: We have corrected the type errors in the manuscript you mentioned. 

The Introduction is acceptable. A short paragraph about salicylic acid signal transduction is followed by a detailed description of several well-known elicitors. The authors could discuss shortly the response of Arabidopsis to H. arabidopsidis to help the reader in the understanding of this mechanism and the chosen parameter which were studied.

Answer: We have added a description of Hyaloperonospora arabidopsidis in line 103.

Please reconsider sentence in lines 85-93, since it is too long. In lines 89-90 “…a series 3,4-dichloroisothiazole-5-carboxamides with systemic acquired resistance were…” is written, but it is unclear what that mean.

Answer: We have corrected the errors in the manuscript you mentioned

Materials and methods must be improved, and the Authors should reconsider the structure of it, and reduce the number of subsections.

Synthesis of LY5-24-2 could be described in one subsection, and Figure 1 legend could provide the reader with slightly more details.

Answer: We have corrected the errors in the manuscript you mentioned.

Cell wall component measurements can be in a single subsection also.

Answer: We have corrected the errors in the manuscript your mentioned. Measurements of cell wall components have been combined in one subsection

Description of lactophenol Trypan Blue (TB) staining is missing.

Answer: The lactophenol trypan blue (TB) staining solution composed of 10 mL phenol, 10 mL glycerol, 10 mL lactic acid, 10 mL H2O2, and 10 mg of Taipan blue dye.  A. thaliana leaves infested with H. arabidopsidis isolate Noco2 were decolorized by chloral hydrate after boiling in TB staining solution for 5 min.  The description of lactophenol Trypan Blue (TB) staining was given in line 186.

Were the Arabidopsis plants really grown on 20°C at day and 22 °C at night?

Answer: We have corrected the errors in the manuscript you mentioned, Arabidopsis plants really grown on 22 °C at day and 20 °C at night. That was given in line 166.

What is the meaning of DMF? Use the whole name of chemicals when first mentioned.

Answer: N,N-Dimethylformamide has been given in line 167

Which statistical method was used to calculate differences? The software and the methods should be given in the Materials and methods.

Answer: Student’s t-test was used to calculate differences.  The software used for the data analysis were GraphPad Prism 8.1 and Statistix 8.1.  The software and the methods were given in line 254.

Plants were sprayed with DMF as mock treatment and with LY5-24-2, ISO and BTH as elicitors. What was the volume of the treatments?

Answer: Twenty seedlings of A. thaliana were sprayed with 100 μM of LY5-24-2, BTH or ISO in 1 mL each. That has been mentioned in line 168.

Please, try to avoid using numbers at the beginning of the sentence, when it is possible.

Answer: We have corrected the errors in the manuscript you mentioned.

Whole name of ascorbate peroxidase and catalase, and EC numbers of the enzymes should be given.

Answer: APX (ascorbate peroxidase APX, EC1.10.3.3) and CAT (catalase, EC 1.11.1.6) have been given in line 24, 25, 245 and 246.

As I mentioned earlier several typos can be found in the manuscript, but especially in Materials and methods. Some examples can be found bellow:

lines 112, 121,132, and 145: dots are unnecessary after subsection titles. What are the meaning of (2) and (3) after the titles in lines 121 and 132?

Answer: We have corrected the errors in the manuscript you mentioned.  The meaning of the compounds (2) and (3) have been given in line 136.

line 153: correct “Zheng et al”

Answer: We have corrected the errors in the manuscript you mentioned.

lines 155-156: correct tag3 and tag7 to tga3 and tga7

Answer: We have corrected the errors in the manuscript you mentioned.

 line 181: correct “… 40 mesh…”

Answer: We have corrected the errors in the manuscript you mentioned.

Please, reconsider sentence in line 201-202

Answer: We have corrected the errors in the manuscript you mentioned.

Results are presented clearly, but some minor changes can be made:

Results from DMF treated plants served as control throughout the manuscript. However, the question arises, is there any effect of the 0.2 % DMF treatment on the plants when compared to non-treated plants? Have the Authors measured any parameters in non-treated plants? It can be added as supplementary material to the manuscript, to demonstrate that there were no significant difference between DMF treated and non-treated plants.

Answer: The maximum DMF content in 100 µM FY5-24-2, ISO or BTH is 0.2%, so 0.2% was chosen as the Mock treatment.  Similar reference was reported by Rodriguez-Salus et al. in UCR (Rodriguez-Salus, M.; Bektas, Y.; Schroeder, M.; Knoth, C.; Eulgem, T. The synthetic elicitor 2-(5-bromo-2-hydroxy-phenyl)-thiazolidine-4-carboxylic acid links plant immunity to hormesis. Plant Physiol. 2016, 170(1): 444. Doi: 10.1104/pp.15.01058.), we validated the systemic acquired resistance in this lab, therefore, we did not test the effects of 0.2% of DMF. Professor Thomas Eulgem use this system for routine research.

It is rather unusual to refer to P-value in the text, it should be avoided.

Answer: We have corrected the errors in the manuscript you mentioned.

Supplementary Tables and Figures, and the description of Supplementary material are missing from the manuscript, it should be added. Figure S7 is mentioned in line 357, however no reference to Figures S3-6 can be found in the manuscript, it should be corrected.

Answer: We have corrected the errors in the manuscript you mentioned. The FigureS3-S6 have been given in line 372.

The Authors use Hpa Noco2 and Hpa. Noco2 abbreviation as well, it should be unified.

Answer: We have corrected the errors in the manuscript you mentioned.

Fig. 2 – scale bars are not enough visible. Check the title of y axes on Fig 4C and 4D.

Answer: We have corrected the errors in the manuscript you mentioned.

Rewrite Figure legends of Figs 2, 3, and 4, and correct the sentence to “Data were shown as the mean of three biological replicates ± SD.” on Figure legends 2, 3, 4, 7, and 8.

Answer: We have corrected the errors in the manuscript you mentioned.

Reference to Figure 3C and Figure 3D should be added and corrected in line 286 and 287, respectively.

Answer: We have corrected the errors in the manuscript you mentioned.

 Discussion is quite short, and did not contain any comparison of the results of this study and the known literature about the well described elicitors ISO and BTH. The results of the newly synthetized elicitor LY5-24-2 and data of other elicitors from the literature can be compared, to emphasize its effectiveness.

Answer: We have compared the results of this manuscript with BTH and ISO in the immune-inducing effects and mechanisms of action in line 443 to 450.

References should be thoroughly checked also. In line 495 Mcdowell should be written as McDowell and the title should be corrected. Reference in line 493 has two authors, and it should be referred in the text as Livak and Schmittgen, 2001

Answer: We have corrected the errors in the manuscript you mentioned.

Reviewer 2 Report

This study synthesized a novel immune elicitor and found the immune induction by this elicitor in Arabidopsis and Cucumber. The overall study is significant however I have the following concerns

Line18 change inducing SAR to SAR inducing

Line 93-95 Not sure what this means

Line 155-6 Shouldn’t this be tga3 and tga7 instead of tag3 and tag7

Line165-6 briefly describe how this was done

Line213 how old were A thaliana seedlings?

Line222 how old were A thaliana seedlings?

In fig 5B why is there only one GO term each in cellular component and molecular function category

Line 327-330 the text doesn’t corelate with Fig5B at all. I see most of the GO term for biological process in Fig5B

Line 330 Why do you say “These suggested that LY5-24-2 might have the most significant effects on genes related to cellular components” when fig5B shows most GO term for biological process.

Lines332-335 how does Fig6 suggest that the DEGs are involved in plant-pathogen interactions

Lines 335-338 please include a list of differentially expressed genes involved in plant-pathogen interaction.

Author Response

Comments and Suggestions for Authors

This study synthesized a novel immune elicitor and found the immune induction by this elicitor in Arabidopsis and Cucumber. The overall study is significant however I have the following concerns

Line18 change inducing SAR to SAR inducing

Answer: We have corrected the errors in the manuscript you mentioned.

Line 93-95 Not sure what this means

Answer: We reorganized the sentences.

Line 155-6 Shouldn’t this be tga3 and tga7 instead of tag3 and tag7

Answer: We have corrected the errors in the manuscript you mentioned.

Line165-6 briefly describe how this was done

Answer: We have corrected the errors in the manuscript you mentioned.

Line213 how old were A thaliana seedlings?

Answer: Two-week old seedlings were used for all analysis in this manuscript, that has been mentioned in line 58.

Line222 how old were A thaliana seedlings?

Answer: Two-week old seedlings were used for all analysis in this manuscript, that has been mentioned in line 158

In fig 5B why is there only one GO term each in cellular component and molecular function category

Answer: LY5-24-2 induced differentially expressed genes were used to GO analysis. GO term in Fig 5B is not all GO term but the part of P vale <0.05.

Line 327-330 the text doesn’t corelate with Fig5B at all. I see most of the GO term for biological process in Fig5B

Answer: We have corrected the errors in the manuscript you mentioned.

Line 330 Why do you say “These suggested that LY5-24-2 might have the most significant effects on genes related to cellular components” when fig5B shows most GO term for biological process.

Answer: We have corrected the errors in the manuscript you mentioned.

Lines332-335 how does Fig6 suggest that the DEGs are involved in plant-pathogen interactions

Answer: The flavonoids, flavonols and anthocyanins produced during the phenylpropanoid biosynthesis and phenylpropanoid metabolism play an important role in plant resistance to pathogens. (Dong, N. Q.; Lin, H. X. Contribution of phenylpropanoid metabolism to plant development and plant–environment interactions FA. J Integr Plant Biol. 2021, 63(1):30.)

Lines 335-338 please include a list of differentially expressed genes involved in plant-pathogen interaction.

Gene ID

Description

KEGG Pathway

AT1G31690

Copper amine oxidase family protein

(ath00950)Isoquinoline alkaloid biosynthesis - Arabidopsis thaliana (thale cress); (ath00360)Phenylalanine metabolism - Arabidopsis thaliana (thale cress); (ath00960)Tropane, piperidine and pyridine alkaloid biosynthesis - Arabidopsis thaliana (thale cress)

AT3G43670

Copper amine oxidase family protein

(ath00950)Isoquinoline alkaloid biosynthesis - Arabidopsis thaliana (thale cress); (ath00360)Phenylalanine metabolism - Arabidopsis thaliana (thale cress);

AT5G43935

flavonol synthase 6

(ath00941)Flavonoid biosynthesis - Arabidopsis thaliana (thale cress); (ath01110)Biosynthesis of secondary metabolites - Arabidopsis thaliana (thale cress);

AT5G63580

flavonol synthase 2

(ath00941)Flavonoid biosynthesis - Arabidopsis thaliana (thale cress); (ath01110)Biosynthesis of secondary metabolites - Arabidopsis thaliana (thale cress);

AT1G30760

FAD-binding Berberine family protein

(ath00940)Phenylpropanoid biosynthesis - Arabidopsis thaliana (thale cress); (ath01110)Biosynthesis of secondary metabolites - Arabidopsis thaliana (thale cress);

AT2G34790

FAD-binding Berberine family protein

(ath00940)Phenylpropanoid biosynthesis - Arabidopsis thaliana (thale cress); (ath01110)Biosynthesis of secondary metabolites - Arabidopsis thaliana (thale cress);

AT3G32980

Peroxidase superfamily protein

(ath00940)Phenylpropanoid biosynthesis - Arabidopsis thaliana (thale cress); (ath01110)Biosynthesis of secondary metabolites - Arabidopsis thaliana (thale cress);

AT5G05340

Peroxidase superfamily protein

(ath00940)Phenylpropanoid biosynthesis - Arabidopsis thaliana (thale cress); (ath01110)Biosynthesis of secondary metabolites - Arabidopsis thaliana (thale cress);

AT5G15180

Peroxidase superfamily protein

(ath00940)Phenylpropanoid biosynthesis - Arabidopsis thaliana (thale cress); (ath01110)Biosynthesis of secondary metabolites - Arabidopsis thaliana (thale cress);

AT5G58400

Peroxidase superfamily protein

(ath00940)Phenylpropanoid biosynthesis - Arabidopsis thaliana (thale cress); (ath01110)Biosynthesis of secondary metabolites - Arabidopsis thaliana (thale cress);

AT3G10340

phenylalanine ammonia-lyase 4

(ath00360)Phenylalanine metabolism - Arabidopsis thaliana (thale cress); (ath00940)Phenylpropanoid biosynthesis - Arabidopsis thaliana (thale cress); (ath01110)Biosynthesis of secondary metabolites - Arabidopsis thaliana (thale cress)

Round 2

Reviewer 1 Report

During the revision of the manuscript (ID: ijms-1700968) the Authors addressed my earlier comments and the manuscript have been improved. I have only a few remarks:

The Authors added EC number of ascorbate peroxidase (APX) and catalase, however in case of APX the EC number (EC 1.10.3.3) is not correct or ascorbate oxidase activity was investigated. APX (EC 1.11.1.11) is oxidizing ascorbate in the presence of H2O2 to monodehydroascorbate. Ascorbate oxidase (EC 1.10.3.3) uses oxygen to catalyze the oxidation of ascorbate to monodehydroascorbate. Please, correct either the EC number or the name and abbreviation of enzyme throughout the manuscript.

In line 580 still “…sssays…” is written.

The units of APX or CAT activities on y axis legend of Figs. 4C, 4D, 8E and 8F could be formatted in the same way, e.g. APX activity (nmol/min/g FW) or CAT activity (nmol/min/g FW).

The manuscript can be accepted after clarifying the above mentioned issues.

Author Response

The Authors added EC number of ascorbate peroxidase (APX) and catalase, however in case of APX the EC number (EC 1.10.3.3) is not correct or ascorbate oxidase activity was investigated. APX (EC 1.11.1.11) is oxidizing ascorbate in the presence of H2O2 to monodehydroascorbate. Ascorbate oxidase (EC 1.10.3.3) uses oxygen to catalyze the oxidation of ascorbate to monodehydroascorbate. Please, correct either the EC number or the name and abbreviation of enzyme throughout the manuscript.

Answer: Thank you for your correction, we have corrected the errors in the manuscript you mentioned in line 25 and 236.

In line 580 still “…sssays…” is written.

Answer: We have corrected the errors in the manuscript you mentioned

The units of APX or CAT activities on y axis legend of Figs. 4C, 4D, 8E and 8F could be formatted in the same way, e.g. APX activity (nmol/min/g FW) or CAT activity (nmol/min/g FW).

Answer: We have corrected the errors in the manuscript you mentioned
